# High-Precision ADC Spectrum Testing under Non-Coherent Sampling Conditions

**DOI:** 10.3390/s22218170

**Published:** 2022-10-25

**Authors:** Xiaofei Peng, Jie Li, Debiao Zhang, Chenjun Hu, Ning Sun, Jie Jiang

**Affiliations:** 1National Key Laboratory for Electronic Measurement Technology, North University of China, Taiyuan 030051, China; 2School of Electronic Information Engineering, Taiyuan University of Science and Technology, Taiyuan 030024, China

**Keywords:** non-coherent sampling, high-precision ADC, spectral leakage, four-parameter sine fitting, parameter estimation

## Abstract

Realizing coherent sampling is one of the major bottlenecks in high-precision ADC spectrum testing. In spectrum analysis, if coherent sampling is not implemented, spectral leakage will result, which in turn leads to inaccurate test results. In this paper, a combined four-parameter sine-curve-fitting algorithm is proposed incorporating non-coherent sampling, with the amplitude, initial phase, and frequency parameters of the sine wave being obtained by fitting. The corresponding coherent sine wave is then calculated and replaced according to the obtained sine wave to reconstruct the new test data, eliminating the requirement of coherent sampling. Numerous simulations demonstrated the functionality and robustness of the algorithm, which was then used to process and analyze the measured data of two commercial high-precision ADCs. The results show that our algorithm can achieve accurate testing of ADC parameters under relaxed test conditions, which verifies the effectiveness and superiority of the scheme.

## 1. Introduction

With the rapid improvement of digital signal processing and the speed of computing, the accompanying increased requirement for system sensitivity has led, in turn, to an increased requirement for the indicators of high-speed and high-precision ADC (Analog to Digital Converter) testing [1,2]. In recent years, ADC spectrum testing has become an area of intense research interest in the mixed-signal IC test field. Of the dynamic parameter test methods available, spectrum analysis is the most commonly used [3]. This method calculates the dynamic parameters of the device to be measured by obtaining the power of the fundamental frequency and all harmonics in the spectrum. However, the challenge with this approach is that the IEEE Standard for Terminology and Test Methods for Analog-to-Digital Converters [4] and IEEE Standard for Terminology and Test Methods of Digital-to-Analog Converter Devices [5] impose several strict requirements, one of which is the implementation of coherent sampling.

The realization of this requirement for coherent sampling in spectrum analysis means that, for example, the input sine wave must be pure enough so that its distortion is much lower than that of the ADC under test. This is a particular challenge when the test environment (such as temperature), the aging of the equipment, and the inability to perform accurate period interception of the signal at the output (i.e., the number of measurement periods is not an integer), lead to skirt effects appearing on the output signal spectrum, as the quantization index corresponding to the fundamental wave will no longer be the unique index. Spectral leakage occurs, and the formula for calculating the spectrum parameters under standard conditions is no longer applicable, so accurate parameters cannot be obtained [6,7].

To suppress spectral leakage due to non-coherent sampling, researchers have proposed several approaches, the most commonly used of which are the windowing technique [8], the interpolated discrete Fourier transform method [9], the fundamental identification and replacement method [10], and the four-parameter sine-fitting method [11].

The windowing technique involves multiplying the digital output of the ADC with a suitable window function to mitigate the leakage of fundamental power. Common window functions include the Hanning, Blackman, Nuttall, and maximum attenuation (MSD) windows [12,13,14,15]. The spectrum of the window function includes a main lobe and multiple side lobes. When using the windowing technique, it is critical to select an appropriate window function for the specific situation, as the use of unsuitable window functions often causes difficulties in testing. A good side lobe performance of the window function means more concentrated power, low harmonics influence, and accurate power calculation, but it also means a wide main lobe and low-frequency resolution. This means that testers need to have a great deal of a priori knowledge of window functions to select the right type to use. At the same time, the industry currently imposes extremely strict restrictions on the selection criteria of the window function, requiring the first side lobe to be lower than the noise floor, which means that most common window functions cannot be applied to high-precision ADCs, while the special windows that meet the requirements are often very complex expressions, which are not conducive to the calculation.

The interpolated discrete Fourier transform method, which uses the fast Fourier transform for parameter estimation by discrete spectral analysis, can effectively solve the spectral leakage problem in some situations [16,17]. Interpolation of spectral lines by adding complementary zeros (zero-filling) after the effective signal in the time domain can make the spectral lines denser and avoid the limitation of frequency resolution, successfully reducing the fence effect to a certain extent. However, the amount of data increases after zero-filling, which leads to longer processing times and poor real-time performance [18].

Researchers have also proposed many solutions to address the difficulties of implementing coherent sampling in high-resolution ADCs during measurements [19,20,21]. Since non-coherent sampling is difficult to avoid, the non-coherent sampling problem is usually studied and analyzed when considering how to relax other strict standard test conditions [22]. For example, when studying input test signals with harmonic distortion, the non-coherent sampling problem is often considered at the same time. The FIRE algorithm proposed by [23,24,25] accomplishes the testing of ADC dynamic parameters under non-coherent sampling conditions by identifying and replacing the fundamental wave of the output signal under non-coherent sampling conditions. However, the Newton iteration part of the FIRE algorithm consumes a large amount of processing time, which is unfavorable for the mass production of ADC chips and will increase the cost of testing.

Another widely used method is the four-parameter sine-fitting method with accurate estimates of spectral parameters [26,27], such as the total harmonic distortion (THD), signal-to-noise ratio (SNR), signal-to-noise and distortion (SINAD), spurious free dynamic range (SFDR), and the effective number of bits (ENOB). However, this approach is limited by its computational inefficiency, the need for pre-estimation of the four fitted parameters, and the need for sufficiently high estimation accuracy [28,29,30]. In addition, the convergence of the four-parameter iterative search method has been problematic. There is no clear conclusion or criterion for judging when it converges, and there is no clear interval of convergence. The closer the initial value is to the optimal value, the more the convergence of the iterative process is guaranteed; it will otherwise be more likely to lead to a divergent iterative process.

All the above methods suffer from one or more problems, such as a large amount of data, long computation time, inaccurate frequency estimation, a lack of robustness in the Nyquist range, a dependence of the results on the selected window function type, or an inability to perform a full-spectrum test. Therefore, there is a need to develop a method that can solve all the above problems and perform spectral tests accurately without the need for coherent sampling.

In this paper, a combined four-parameter sine-fitting algorithm is proposed for the theoretical derivation of the spectral leakage caused by non-coherent sampling. First, a three-parameter sine fit is performed using the input signal frequency as the known frequency to obtain the four parameters of amplitude, frequency, phase, and DC component. Then, using this as the initial value, the proposed four-parameter sine fit algorithm is used without iteration and absolute convergence to obtain the real test signal. Finally, the corresponding coherent sine wave is calculated and replaced according to the obtained test signal to reconstruct the new test data, eliminating the influence of spectral leakage of the fundamental on the spectrum test. The proposed algorithm is verified by simulation and experiment, and, finally, the accurate testing of high-precision ADC dynamic parameters is achieved under non-coherent sampling test conditions.

The rest of the paper is divided into the following sections. Section 2 discusses the fundamentals of the ADC spectrum testing method. Section 3 presents the proposed algorithm. Section 4 describes the simulation and robustness verification of the algorithm. Section 5 verifies the applicability of the algorithm to determining ADC performance using a test board. Section 6 concludes the paper.

## 2. ADC Spectrum Testing

The basic principle of spectrum analysis is to input a high-purity sine signal to the ADC under test and then acquire the output quantized digital signal and perform a fast Fourier transform on this data to obtain the spectrum of the ADC to calculate its relevant dynamic parameters.

If we let x(t) be the time-domain expression of the simulated input signal, modeled with harmonic distortion, then the time-domain expression is:
(1)$$x(t)=A\cos\left(2\pi f_{sig}t+\varphi \right)+\sum_{h=2}^{H}A_{h}\cos\left(2\pi hf_{sig}t+\varphi_{h}\right)+w\left(t\right)$$
where A is the amplitude of the signal x(t), $f_{sig}$ is the input signal frequency, φ is the initial phase of the signal, H is the total number of harmonics, $A_{h}$ and $\varphi_{h}$ are the amplitude and initial phase of the harmonics, $A_{h}$ << A for all values of *h* taken 2 ≤ h ≤ H, whose initial phase $\varphi_{h}$ is distributed in the interval [0, 2π], and w is the noise in the signal.

If we let x[n] be the analog interpretation of the digital output obtained from an ADC whose gain error and offset error have been calibrated, it can be expressed as:(2)$$x\left[n\right]=A\cos\left(\frac{2\pi J}{M}n+\varphi \right)+\sum_{h=2}^{H}A_{h}\cos\left(\frac{2\pi hJ}{M}n+\varphi_{h}\right)+w\left[n\right]$$
where n = 0, 1, 2, …, M − 1, M is the total number of sampling points, generally taken to the power of 2, J is the number of periods of the intercepted sampled signal, also known as the number of measurement periods, and w[n] is the noise at the nth sample point. Harmonics in the ADC output signal represent a nonlinear distortion of the ADC [21]. Discrete Fourier transform (DFT) with M sampling points is performed on the sampling sequence to obtain the exact spectral parameters. The expression for the Fourier transform of x[n] is:(3)$$X_{k}=\frac{1}{M}\sum_{n=0}^{M-1}x\left[n\right]e^{-j\frac{2\pi k}{M}n},k=0,1,\cdots ,M-1$$
where k is the index of the frequency code value.

Coherent sampling requires that the sampling time be an integer multiple of the input signal period [22], which has the advantage of avoiding spectral leakage. The international ADC test standard defines coherent sampling as:(4)$$\frac{f_{sig}}{f_{samp}}=\frac{J}{M}$$
where $f_{samp}$ is the sampling frequency when the number of measurement periods J is an integer and the total number of sampling points M is coprime with J. To avoid repeated sampling, the same point is taken at different periods to achieve coherent sampling.

Ideally, information on the fundamental frequency and all harmonics can be obtained in the spectrum of the ADC to calculate various dynamic parameters such as SNR, SFDR, or THD [24]. However, the sampling process of the input signal requires the input signal frequency to be accurate to 8–12 decimal places, and this condition is difficult to achieve, resulting in the coherent sampling condition not being met and spectral leakage, which will lead to the leakage of fundamental power to nearby frequencies, affecting or even drowning the harmonics, thus, preventing the accurate calculation of the fundamental frequency and all harmonics, and, ultimately, a failure to obtain the correct parameter test results. Figure 1 shows the spectrum obtained under coherent and non-coherent sampling conditions.

## 3. Fundamental Waveform Estimation and Reconstruction Algorithm

Before describing our new approach to spectral analysis under non-coherent sampling conditions in detail, we will briefly describe the fundamental frequency estimation and reconstruction methods [22,23].

When DFT is applied to test data under non-coherent conditions, the leakage in the spectrum is mainly due to leakage of the fundamental frequency. For high-resolution ADC tests, the leakage at all frequencies other than the fundamental frequency is significantly lower than the total noise power of the ADC. This effect is illustrated in Figure 2 and Figure 3.

Figure 2 shows the DFT spectrum of the non-coherent test data. There is significant spectral leakage around the fundamental. However, if the non-coherent fundamental frequency in this data is estimated and removed, accurate harmonic and noise information can be obtained from the remaining spectrum (Figure 3). This demonstrates that an accurate estimation of the non-coherent fundamental frequency is required to obtain the correct spectral results.

### 3.1. Fitting of Sine Test Signals

There are several methods for estimating non-coherent fundamental waves, such as the interpolated discrete Fourier transform method [16], which first adds a window to the non-coherent sampled data and then interpolates to accurately estimate the fundamental and the previously-discussed FIRE algorithm [24], with its time-intensive Newton iteration. However, we propose a combined four-parameter sine-fitting algorithm, which first uses the frequency of the input signal as the known frequency as the basis of three-parameter sine fitting to obtain the amplitude, initial phase, DC component, and fitting residuals of the fundamental frequency and then combines frequency estimation and three-parameter sine fitting to implement a four-parameter least-squares fitting algorithm to finally obtain an accurate test signal.

#### 3.1.1. Three-Parameter Sine-Fitting Method

The input signal is modeled in the same way as Equation (1), and the input signal expression is obtained by splitting Equation (1) using the sum and difference product formula in trigonometric functions as follows:(5)$$x\left(t\right)=A_{0}\cos\left(2\pi f_{sig}t+\varphi \right)+c_{0}=a_{0}\cos\left(2\pi f_{sig}t\right)+b_{0}\sin\left(2\pi f_{sig}t\right)+c_{0}$$
where $a_{0}=A_{0}\cos(\varphi )$, $b_{0}=-A_{0}\sin(\varphi )$, and $c_{0}$ are the DC components. The equal interval uniform data acquisition sequence is $x_{1},x_{2},\cdots ,x_{n}$. If the sampling interval is taken as Δt, the acquisition rate v, the acquisition time $t_{i}=i\times \Delta t=i/v$ of the sampling point $x_{i}$, i=1,2,⋯,n, and the digital angular frequency $\omega =2\pi f_{sig}/v$, then the signal can be expressed in the following discrete form:(6)$$x\left(i\right)=A_{0}\cos\left(\omega i+\varphi_{0}\right)+c_{0}=a_{0}\cos\left(\omega i\right)+b_{0}\sin\left(\omega i\right)+c_{0}$$

The three-parameter sine-curve-fitting process, using the frequency of the input signal as the initial value, minimizes the sum of squares of the residuals ε described in Equation (6) by searching for a, b, and c:(7)$$\varepsilon =\overset{n}{\underset{i=1}{\sum }}{\left[x_{i}-a\cos\left(\omega i\right)-b\sin\left(\omega i\right)-c\right]}^{2}$$

Then a, b, and c are the least squares fits of $a_{0}$, $b_{0}$, and $c_{0}$.

The fitted function is obtained when the value ε reaches its minimum:(8)$$\hat{x}\left(i\right)=\mathrm{acos}\left(\omega i\right)+\mathrm{bsin}\left(\omega i\right)+c$$

That is:(9)$$\hat{x}\left(i\right)=A\cos\left(\omega i+\theta \right)+c$$
(10)$$A=\sqrt{a^{2}+b^{2}}$$
(11)$$\theta =\left\{ \begin{array}{l} \arctan\left(\frac{-b}{a}\right),a\geq 0\\ \arctan\left(\frac{-b}{a}\right)+\pi ,a<0 \end{array}\right.$$

The fitted residuals RMS value can be written:(12)$$\rho =\sqrt{\frac{\varepsilon }{n}},\;\varepsilon =\overset{n}{\underset{i=1}{\sum }}{\left(x_{i}-\hat{x}\left(i\right)\right)}^{2}$$

As this is a closed algorithm, there is no convergence problem.

#### 3.1.2. Four-Parameter Combination Estimation Algorithm

The input signal frequency estimation is combined with the above three-parameter sine-curve-fitting method to obtain the estimation results of the four parameters of the sine waveform. However, when the sampling interval is sufficiently small, the difference between the amplitude values of adjacent sampling points will be negligible. When the difference is less than a minimum amplitude quantization step, adjacent analog value points may have the same quantization step code value. Therefore, the difference between the cumulative formula of the difference between adjacent sampling points and the theoretical value becomes too large, leading to the error of the obtained digital angular frequency ω becoming too large, which affects the accuracy of other parameters obtained by the three-parameter fitting algorithm based on ω. Here, we propose to replace the subtraction of adjacent sampling points with the subtraction of two adjacent sampling points of k, to avoid the influence of too many adjacent sampling points with the same quantization code value and, thus, form the following improved algorithm.

The sampling sequence of the sine signal waveform $x_{1},x_{2},\cdots ,x_{n}$, whose function relationship is shown in Equation (6) can be written:(13)$$z\left(i\right)=A_{0}\cos\left(\omega i+\varphi_{0}\right)\;i=1,2,\cdots ,n$$

If we let the error of the measurement point $x_{i}$ be $\gamma_{i}$, and q=2cosω, then:(14)$$x_{i}=x\left(i\right)+\gamma_{i}=z\left(i\right)+c+\gamma_{i}$$
(15)z(i)+z(i−2)=(2cosω)⋅z(i−1)=q⋅z(i−1)
(16)x(i)−c+x(i−2)−c=q⋅(x(i−1)−c)
(17)$$x_{i}-c-\gamma_{i}+x_{i-2}-c-\gamma_{i-2}=q\cdot (x_{i-1}-c-\gamma_{i-1})$$
(18)$$x_{i+1}-c-\gamma_{i+1}+x_{i-1}-c-\gamma_{i-1}=q\cdot (x_{i}-c-\gamma_{i})$$
which we can rearrange into:(19)$$\lambda_{k,i}=x_{i+k}-x_{i}$$
(20)$$\eta_{k,i}=\gamma_{i+k}-\gamma_{i}$$
(21)$$x_{i+k+1}-c-\gamma_{i+k+1}+x_{i+k-1}-c-\gamma_{i+k-1}=q_{k}\cdot \left(x_{i+k}-c-\gamma_{i+k}\right)$$
(22)$$\lambda_{k,i+1}+\lambda_{k,i-1}-q_{k}\cdot \lambda_{k,i}=\eta_{k,i+1}+\eta_{k,i-1}-q_{k}\cdot \eta_{k,i}$$
where k is a natural number ranging from one to N, and N is the number of samples in each sine wave period. The accuracy of the estimated parameters changes periodically as the value of k increases, and the change period is the number of sampling points per cycle N. The best estimation results are obtained when k≅N/2. In this case, the accuracy of the estimated values of each parameter is the highest. Here, $q_{k}=2\cos\omega_{k}$ and $\eta_{k,i}$ is the random error, which is obtained by selecting $q_{k}$ such that:(23)$$\rho_{k}=\overset{n-k-1}{\underset{i=2}{\sum }}{\left(\lambda_{k,i+1}+\lambda_{k,i-1}-q_{k}\cdot \lambda_{k,i}\right)}^{2}=min$$

Let $\frac{d\rho_{k}}{dq_{k}}=0$, then we obtain:(24)$$q_{k}=\frac{\lambda_{k,n-k-1}\lambda_{k,n-k}+\lambda_{k,1}\lambda_{k,2}+2\sum_{i=2}^{n-k-2}\lambda_{k,i}\lambda_{k,i+1}}{\sum_{i=2}^{n-k-2}\lambda_{k,i}^{2}}$$
(25)$$\omega_{k}=\arccos\left(\frac{q_{k}}{2}\right)$$
$\omega_{k}$ is the least square estimation of the digital angular frequency of sine wave, and the real input signal frequency f can be obtained from $\omega_{k}=2\pi f/v$. After using the three-parameter estimation method, the estimated values of the other three parameters can be obtained as A for the amplitude, φ for the initial phase, and c for the DC component, thus, completing the four-parameter estimation. The resulting test signal obtained by fitting can then be:(26)$$x_{t}=A\cos\left(2\pi ft+\varphi \right)+c$$

### 3.2. Reconstructing Coherent Sine Signals

To obtain coherent test data, the test signal obtained by fitting is replaced in the test data by the reconstructed coherent test signal, and then the reconstructed test data is subjected to DFT to obtain accurate spectral test results [20]. The test signal expression, which is derived from Equation (4), can be written:(27)$$x(t)=A\cos\left(2\pi \frac{J+\delta }{M}f_{samp}t+\varphi \right)+c$$
where δ = 0 for coherent sampling, and δ ≠ 0 for non-coherent sampling. As Equation (26) is the test signal obtained by fitting, the frequencies in Equations (26) and (27) are equal, and it is obtained that:(28)$$f=\frac{J+\delta }{M}f_{samp}$$

From Equation (28), we obtain J+δ as:(29)$$J+\delta =M\frac{f}{f_{samp}}$$
where f is the true frequency obtained from the test data. To ensure that all samples are valid and there are no redundant samples, the total number of sample points M is mutually prime with the number of periods J, and the length of the test data at the time of testing M is an integer power of two. When this is the case, any odd value of J satisfies the mutual prime. Reassigning the nearest odd value of J+δ to $J_{new}$, the signal frequency of the coherent test condition is obtained from $J_{new}$:(30)$$f_{r}=\frac{J_{new}}{M}f_{samp}$$

Replacing the frequency f in Equation (26) with $f_{r}$ yield:(31)$$x_{r}=A\cos\left(2\pi f_{r}t+\varphi \right)+c$$
where $x_{r}$ is the coherent test signal, and the non-coherent sampling fundamental signal is removed from the original test data and replaced with a coherent sampling fundamental signal by Equations (26) and (31) to obtain the new test data $x_{new}\left[n\right]$:(32)$$x_{new}\left[n\right]=x\left[n\right]-x_{t}\left[n\right]+x_{r}\left[n\right]$$

DFT of the test data provides the frequency domain information $x_{\mathrm{new}k}$, which in turn yields accurate ADC dynamic parameters [31]. The algorithm can perform spectrum testing without windowing and requires no iterations, absolute convergence, or high computational speed, which saves time and reduces testing costs. The flowchart in Figure 4 shows the process flow of spectrum testing using a combined four-parameter sine-curve-fitting algorithm.

## 4. Numerical Simulation

In this section, effective simulation experiments are performed to verify the functionality and robustness of the proposed algorithm. The computation time of this algorithm is also compared with other non-coherent sampling algorithms in the literature.

### 4.1. Functionality

Firstly, the output signal of a 16-bit ADC is simulated under the conditions of non-coherent sampling and harmonic distortion of the input signal. The frequency of the input signal is 50 kHz, the sampling rate is 480 kS/s, and the total number of sampling points is 32,768. The spectrum of the original output signal, the coherently sampled signal, and the signal after processing by the proposed algorithm are shown in Figure 5. The test signal is a single frequency signal source. According to the actual test situation, the amplitude of harmonics and noise has been large enough. In the real test situation, the harmonics and noise generated by the power supply and the peripheral circuit of the system are far smaller than those set in the text. The simulation conditions have been set enough to cover the actual application test.

For the above signal, the number of measurement periods calculated by Equation (4) is 3413.33, i.e., a non-integer number; therefore, the original output signal is obtained under non-coherent sampling conditions; thus, spectral leakage will occur in the spectrum, which cannot be directly used to calculate the dynamic parameters by the standard calculation formula, as shown in the blue curve in Figure 5.

Processing the non-coherent sampling data represented by the blue curve with the proposed algorithm results in the green curve. A comparison of the green and blue curves shows that the use of the algorithm eliminates the spectral leakage caused by non-coherent sampling completely, and the clean spectrum has the same effect as the output signal spectrum of the coherent test signal (red curve) under ideal test conditions, which verifies the effectiveness of the algorithm.

### 4.2. Robustness

To verify the reliability of the proposed algorithm under different degrees of non-coherence, we performed many random simulations. As a reference, the parameters obtained from a set of coherently sampled data were used as the real values. The fractional part of the sampling period delta was randomly generated in the range of –0.5 to 0.5, and all other simulation settings were the same as in the functional verification. Further, 1000 simulations were performed. The non-coherent data from each simulation were processed using our algorithm. The error between the obtained results and the real values for the 1000 simulations is shown in Figure 6. After processing with our algorithm, the maximum error of each parameter is less than 2 dB in absolute value, and the algorithm can still obtain accurate results even with severe non-coherent sampling (when the delta is very close to 0.5). This test confirms that, for any degree of non-coherent sampling, the errors of each parameter are within the acceptable range, and the algorithm is reliable.

### 4.3. Computation Time

In Table 1, we list the computation time of the direct DFT using different window functions and the FIRE algorithm. Among all the methods listed, the algorithm proposed in this paper provides accurate test results with the least cost in terms of computation time. It can also be seen from the table that only one window function can accurately test the 16-bit ADC. This demonstrates the degree to which the use of the window function for spectrum testing depends on prior knowledge of the ADC resolution, while the special windows that meet the requirements are often very complex expressions, which are not conducive to computation. The FIRE algorithm provides accurate test results but takes a long time to perform the Newton iteration, which is not conducive to mass production testing of ADC chips.

## 5. Experimental Verification

To further demonstrate the utility of the proposed algorithm, we conducted a spectral test of non-coherent sampling by using the proposed algorithm for two commercial high-precision ADCs. A Kintex-7-FPGA based A/D converter test system was designed and built for testing the dynamic parameters of the ADCs, with the overall scheme shown in Figure 7. 

The test evaluation board used to test the performance of the ADC is shown in Figure 8 below. It uses a high-speed Kintex-7 FPGA as the core device, which carries out the sampling of the output data of the high-performance A/D converter. It also allows the caching of data and the processing of data across clock domains. DDR3 SDRAM is used as the memory for the sampling data during A/D conversion, with a data throughput rate of up to 20 Gbits/s, which meets the requirements of high bandwidth, high storage depth, and high-speed real-time data reading and writing for high-performance testing. The SDRAM uses EMMC to realize high speed and large capacity storage, with a total effective storage capacity of 2TB, four pieces of storage speed up to 250MB/s*4 = 1GB/s. The board is completed by ethernet and fiber optic interface protocol implementation and data transmission. In addition, the JESD204B standard transceiver interface has been integrated into the FPGA, which provides great convenience for data sampling and the sending of high-performance A/D converters.

### 5.1. AD976 Test

The proposed algorithm was first tested using a 16-bit high-precision AD976. The procedure for the test was as follows:Connect the AD976 board to be tested to the DC power supply and the evaluation board in the specified environment; input a high-precision sine wave $f_{in}=44,055.15234375$ Hz with the sampling frequency set to $f_{samp}=200$ kHz.Power up the device and mode control, and connect the device’s digital output to the digital acquisition terminal through the high-speed interface.Provide a frequency-specific analog input signal through a high-performance RF signal source, and connect a fixed-frequency filter to the AD976 analog input.Use the logic analyzer/evaluation board to set the AD976 for dynamic conversion and acquisition of the digital output signals of the device.DFT to the obtained data to obtain the frequency domain information.

As expected, there was significant spectral leakage in the non-coherently sampled data. However, applying our algorithm, clean and accurate spectra were obtained, as is shown by the red spectrum in Figure 9, which matches the blue spectrum (coherent) and provides accurate spectral results for the high-resolution ADC. To show the effect of the use of a window function, the black plot in Figure 9 shows the spectrum obtained when using the Window in [13] function for the same non-coherent sampled data. Table 2 compares the SNR, SINAD, THD, and SFDR values of the ADC using our algorithm and the windowing technique with the values obtained using a coherent sampling method. The algorithm accurately estimates the parameters and suppresses the spectral leakage somewhat better than the Window in [13] function, and the higher the number of ADC bits, the better the suppression of leakage.

### 5.2. AD9230 Test

To further verify the proposed algorithm, another commercial high-precision ADC was also tested (Figure 10). The procedure, in this case, was as follows.

Apply a 1.8 V power supply.Apply a voltage of −1 dBFS amplitude to the ADC under test, then a sine wave source at $f_{in}=10,002,700$ Hz, and filter the sine wave input to remove distortion and random noise from the input signal.Apply a sample clock with the specified sampling frequency $f_{samp}=75,366,400$ Hz to the ADC under test.After the ADC has stabilized, collect 32,768 output conversion data points.Use the proposed algorithm and DFT to obtain the spectrum.

The obtained spectrum is shown in Figure 11. Our algorithm successfully eliminated the spectral leakage. Table 3 shows the test results, which demonstrate the accuracy of the proposed algorithm, achieving accurate estimation of the dynamic parameters SNR, SINAD, and ENOB.

In summary, our proposed algorithm can accurately identify and reconstruct the fundamental frequency of the output signal under non-coherent sampling and input signal with harmonic distortion test conditions and achieve accurate testing of the ADC dynamic parameters.

## 6. Conclusions

Under the test conditions of non-coherent sampling and an input signal with harmonic distortion, the ADC dynamic parameters cannot be calculated directly due to spectral leakage. In this paper, a combined four-parameter sine-curve-fitting algorithm is proposed to obtain the amplitude, initial phase, and frequency parameters of the sine wave by fitting. This is used to construct a coherent sampling signal of the fundamental frequency that replaces the non-coherent sampling signal of the fundamental frequency in the test data, and the new signal obtained greatly improved the spectral leakage. Unlike windowing, this algorithm can be used to test any ADC output without prior knowledge of the ADC resolution. The accuracy and robustness of the proposed algorithm were verified by numerical simulation, and a simulation-based computation time comparison verifies the operational efficiency of the algorithm. Finally, two commercial high-precision ADCs, the AD976 and the AD9230, were evaluated, and the results were SNR: 74.321 dB, SINAD: 73.654 dB, SFDR: 79 dB, and THD: 89.024 dB for the AD976, and SNR: 62.329 dB, SINAD: 63.097 dB, and ENOB: 9.198 bits for the AD9230. These results show the practical effectiveness of the proposed algorithm. The algorithm can achieve accurate testing of ADC dynamic parameters under arbitrary non-coherent sampling conditions without complex calculations or tedious manual calibration, greatly reducing test time and test cost.

## Figures and Tables

**Figure 1 sensors-22-08170-f001:**
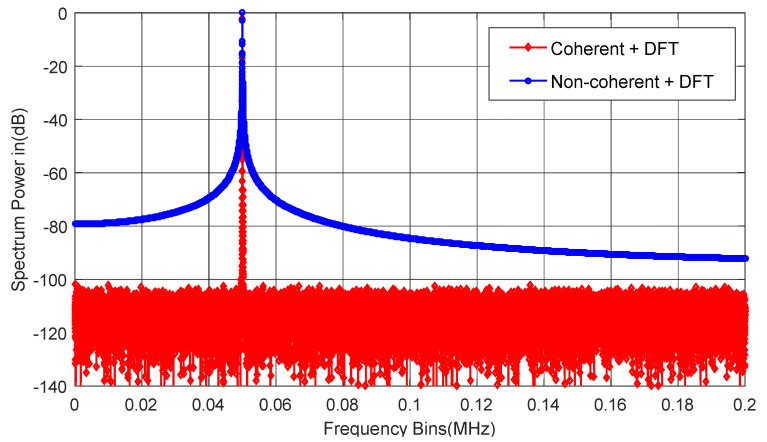
Coherent (red) and non-coherent (blue) sampling spectra.

**Figure 2 sensors-22-08170-f002:**
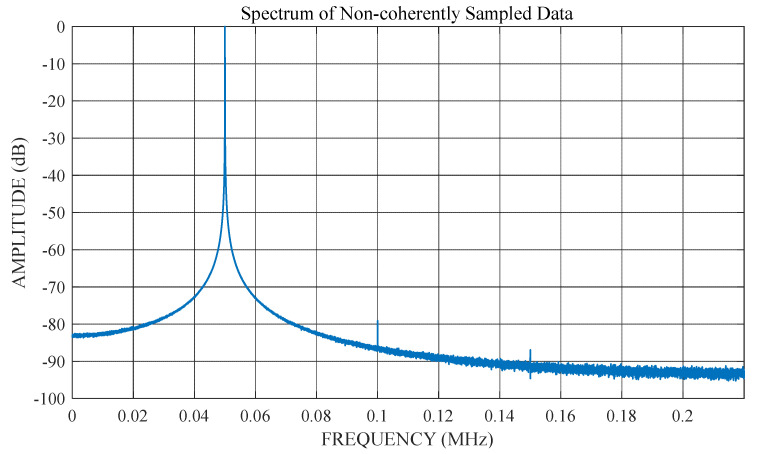
Spectral leakage in the DFT of the data for non-coherent sampling.

**Figure 3 sensors-22-08170-f003:**
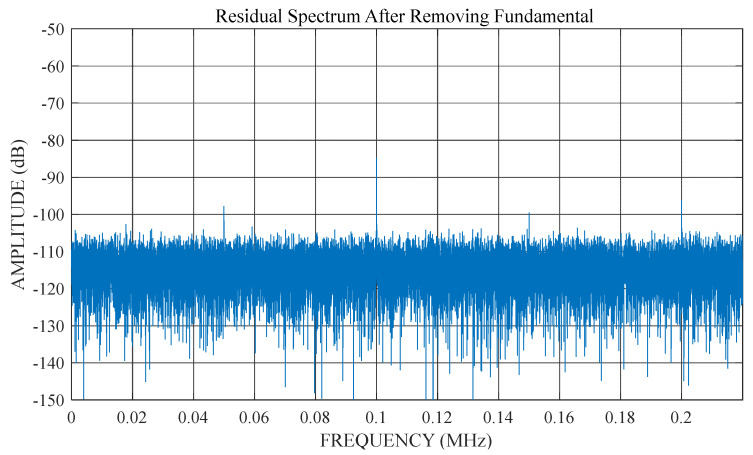
Residual spectrum after removing the fundamental frequency.

**Figure 4 sensors-22-08170-f004:**
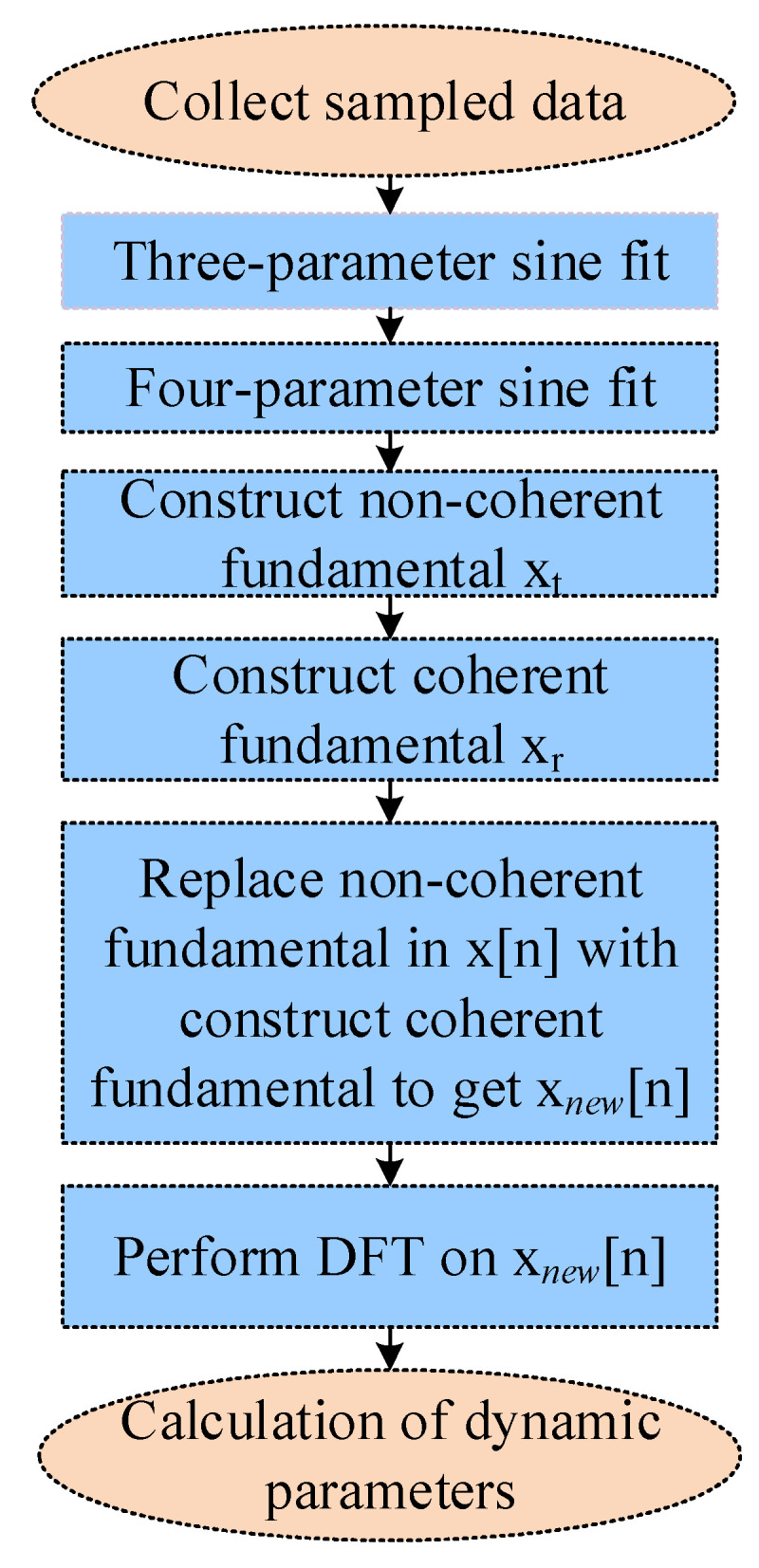
Algorithm process flow.

**Figure 5 sensors-22-08170-f005:**
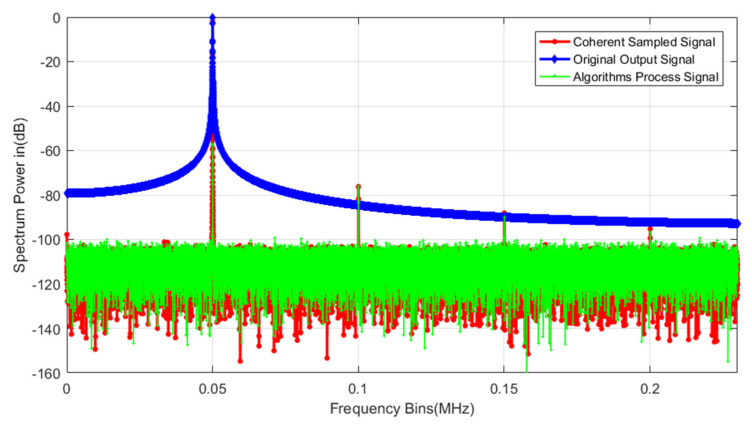
The simulated spectrum of a 16-bit ADC under three sets of conditions (original, coherently sampled, and after processing with the proposed algorithm).

**Figure 6 sensors-22-08170-f006:**
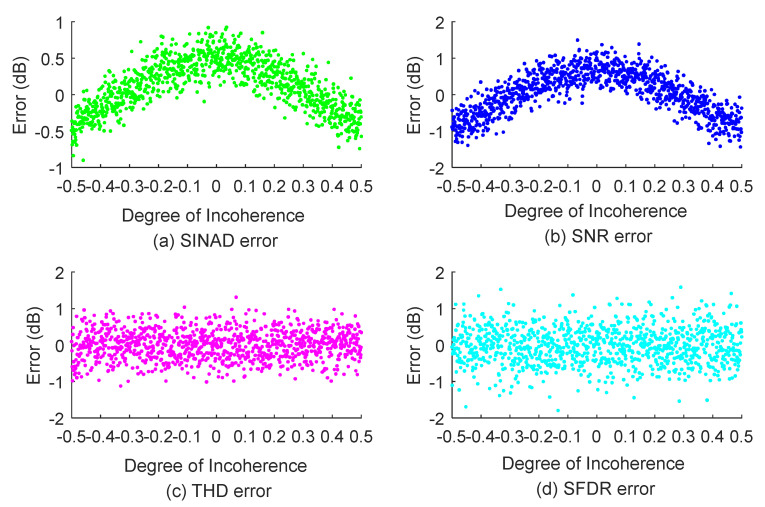
Algorithm robustness verification.

**Figure 7 sensors-22-08170-f007:**
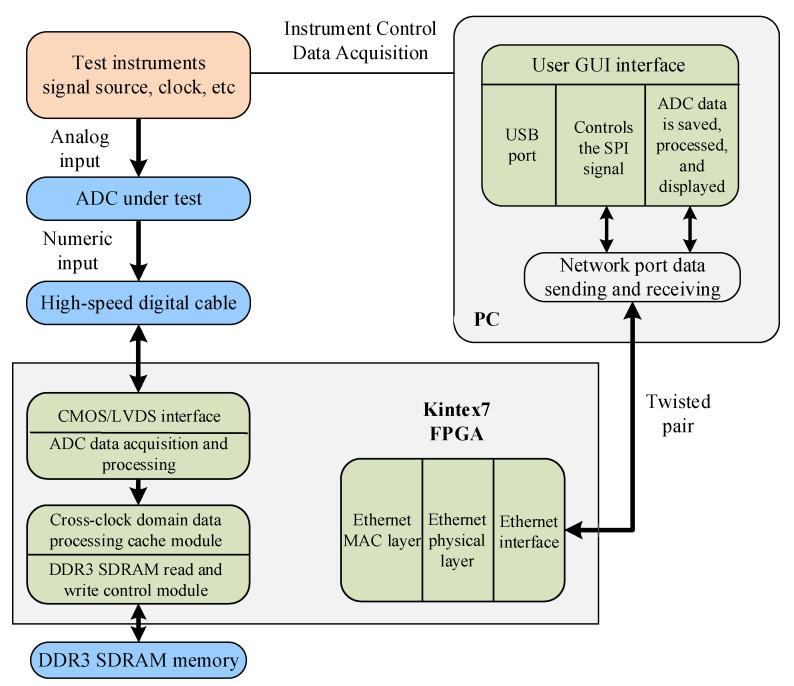
Overall scheme of the ADC spectrum testing setup.

**Figure 8 sensors-22-08170-f008:**
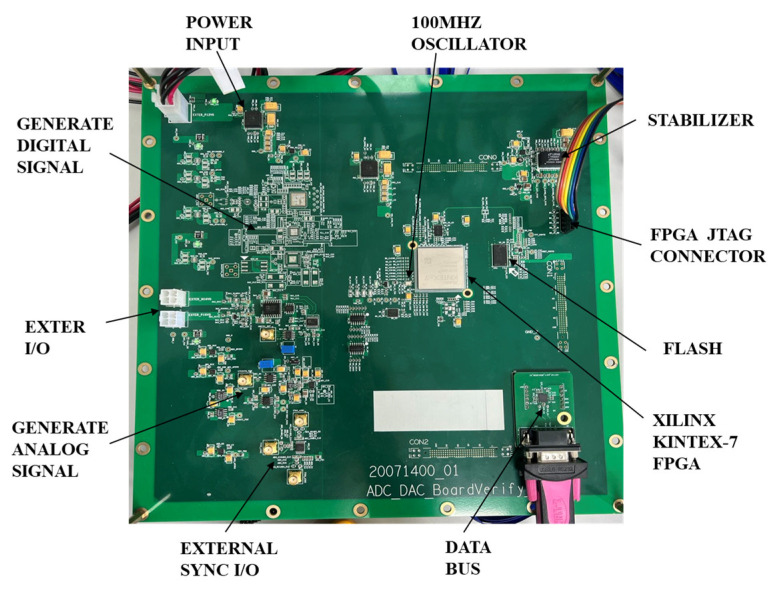
Test evaluation board.

**Figure 9 sensors-22-08170-f009:**
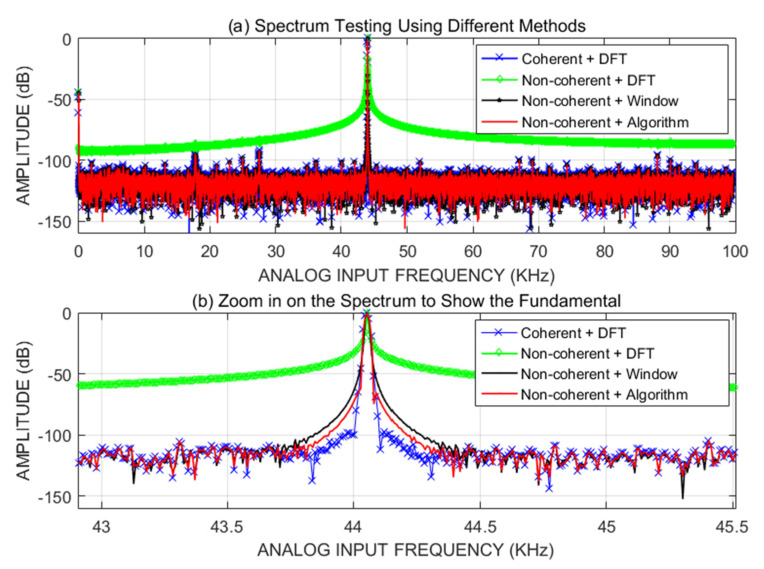
Spectrum tests using different test methods: (**a**) full spectrum; and (**b**) amplified fundamental spectrum [13].

**Figure 10 sensors-22-08170-f010:**
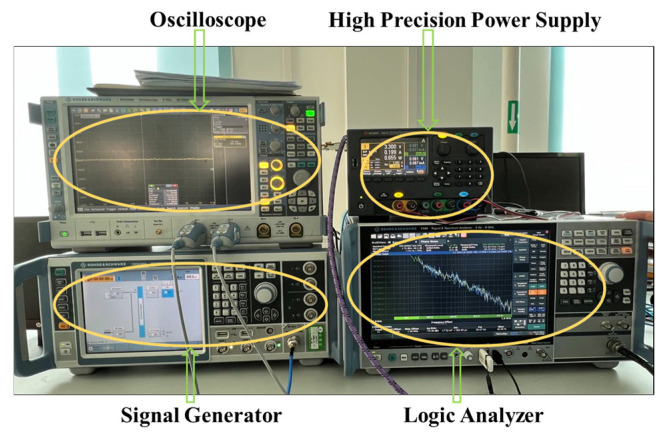
Test equipment for the AD9230 test.

**Figure 11 sensors-22-08170-f011:**
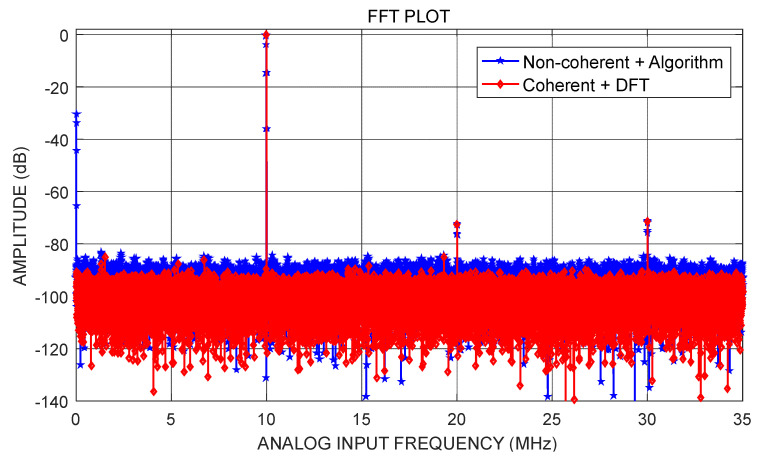
Spectrogram of AD9230 test data.

**Table 1 sensors-22-08170-t001:** Computation time and functionality comparison. (J = 3413.33, M = 32,768, 16-b ADC).

Method	Time (ms)	Functionality
Direct DFT	39	Inaccurate
Hanning	40	Inaccurate
Blackman-Nuttall (4-term)	42	Inaccurate
Window in [13]	87	Accurate
FIRE	80	Accurate
**Proposed Method**	**72**	**Accurate**

**Table 2 sensors-22-08170-t002:** Dynamic test results.

Date Set	Method	SNR (dB)	SINAD (dB)	THD (dB)	SFDR (dB)
Test 1	Coherent	75.065	75.145	−90.248	81.142
	**Non-coherent + Algorithm**	**74.112**	**73.373**	**−88.566**	**79.523**
	Non-coherent + Window in [13]	73.553	72.469	−87.935	77.946
Test 2	Coherent	75.213	75.314	−90.568	80.997
	**Non-coherent + Algorithm**	**74.248**	**73.674**	**−89.023**	**79.653**
	Non-coherent + Window in [13]	73.973	71.466	−88.575	78.795
Test 3	Coherent	74.965	75.665	−89.844	81.616
	**Non-coherent + Algorithm**	**73.568**	**74.472**	**−88.265**	**79.945**
	Non-coherent + Window in [13]	72.761	72.429	−86.963	78.149

**Table 3 sensors-22-08170-t003:** Results of the AD9320 test.

Dynamic Indicators	Standard Values in Datasheet	Test Results
SNR	63.8 dB (Min); 64.6 dB (Typ);	62.329 dB
SINAD	63.7 dB (Min); 64.5 dB (Typ);	63.097 dB
ENOB	10.6 bits (Typ)	9.198 bits
WORST HARMONIC (Second OR Third)	−82 dB (Typ); −78 dB (Max);	−78.280 dB
WORST OTHER (SFDR Excluding Second and Third)	−89 dB (Typ); −84 dB (Max);	−85.247 dB

## Data Availability

Not applicable.

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
