# Peer review of "High-Precision ADC Spectrum Testing under Non-Coherent Sampling Conditions"

_sensors, 2022, doi:10.3390/s22218170_

Round 1
Reviewer 1 Report
This paper presents an approach of spectrum analysis under non-coherent sampling. Using the proposed approach. From the test result, it has very good performance.
Whether the authors use MATLAB to process the data? Can you share the code and use an example in the code to show the result?
Author Response
First of all, I would like to thank the reviewers for their meticulous review of this article in their busy schedules, thank you for your affirmation of our work, and sincerely thank you for your advice.
In this paper, the data is processed by MATLAB. Since the research is undertaken in the military chip application verification project, the code and data used in the data processing are classified files, so we are deeply sorry for the inconvenience caused to you because they cannot be opened.
Reviewer 2 Report
Eq. (23) is a bit strange, not a formal expression.
Eq. (25) is the same as Eq. (26), it should be a typo, please correct it.
In the test of the 16-bit high-precision AD976, the SNR, SINAD, THD, and SFDR values in Table 2 all failed to reach the 16-bit specification. what is the reason?
In the test of AD9320, the test results in Table 3 cannot be consistent with its DataSheet. There are about 2 dB drops. What is the reason?
Reviewer 3 Report
The authors propose a combined four-parameter sine curve fitting algorithm incorporating non-coherent sampling. The corresponding coherent sine wave is then calculated and replaced according to the obtained sine wave to reconstruct the new test data.
a) What is the level of noise used in the studied signals?
b) What is the maximum frequency for the input signal?
c) The time computation for the FIRE algorithm is not a long time compared with the proposed method; it is just 8 ms of difference, maybe it is right that mass production testing of ADC chips could be high, but is any way to calculate this computation time?
d) Could this methodology be applied for a spectrum on time-frequency, for example, in the case where the frequency changes with the time (chirp signal)?
e) It is necessary to another table with the comparison of your proposed method with different levels of noise also applied to the other compared methods.
f) It was not clear for this reviewer the combined four-parameter sine curve fitting algorithm.
Round 2
Reviewer 2 Report
Based on Table 2, the proposed method still has room for improvement compared to the coherent method.